

# Freshwater sponge hosts and their green algae symbionts: a tractable model to understand intracellular symbiosis

Chelsea Hall[1,2], Sara Camilli[1,3], Henry Dwaah[1], Benjamin Kornegay[1], Christie Lacy[1], Malcolm S. Hill[1,4] and April L. Hill[1,4]

[1] Biology, University of Richmond, Richmond, VA, United States of America
[2] Department of Microbiology, Immunology, and Cancer Biology, University of Virginia School of Medicine, Charlottesville, VA, USA
[3] Lewis-Sigler Institute for Integrative Genomics, Princeton University, Princeton, NJ, United States of America
[4] Biology, Bates College, Lewiston, ME, United States of America

## ABSTRACT

In many freshwater habitats, green algae form intracellular symbioses with a variety of heterotrophic host taxa including several species of freshwater sponge. These sponges perform important ecological roles in their habitats, and the poriferan:green algae partnerships offers unique opportunities to study the evolutionary origins and ecological persistence of endosymbioses. We examined the association between *Ephydatia muelleri* and its chlorophyte partner to identify features of host cellular and genetic responses to the presence of intracellular algal partners. *Chlorella*-like green algal symbionts were isolated from field-collected adult *E. muelleri* tissue harboring algae. The sponge-derived algae were successfully cultured and subsequently used to reinfect aposymbiotic *E. muelleri* tissue. We used confocal microscopy to follow the fate of the sponge-derived algae after inoculating algae-free *E. muelleri* grown from gemmules to show temporal patterns of symbiont location within host tissue. We also infected aposymbiotic *E. muelleri* with sponge-derived algae, and performed RNASeq to study differential expression patterns in the host relative to symbiotic states. We compare and contrast our findings with work in other systems (e.g., endosymbiotic *Hydra*) to explore possible conserved evolutionary pathways that may lead to stable mutualistic endosymbioses. Our work demonstrates that freshwater sponges offer many tractable qualities to study features of intracellular occupancy and thus meet criteria desired for a model system.

Corresponding author
April L. Hill, ahill5@bates.edu

# INTRODUCTION

A watershed moment for life on this planet involved the successful invasion of, and persistent residence within, host cells by bacterial symbionts (i.e., proto-mitochondria and proto-chloroplasts), which opened evolutionary pathways for multicellular organisms (*Margulis, 1993*). Indeed, endosymbioses that involve benefits for both interacting partners are abundant in modern ecosystems (*Douglas, 2010*; *Bordenstein & Theis, 2015*).

Intracellular symbioses involving phototrophic symbionts and heterotrophic hosts are particularly important given that they support many ecological communities. For example, populations of Symbiodiniaceae harbored by cnidarian and other invertebrate hosts energetically subsidize the entire coral reef ecosystem (*Stambler, 2011*). In many freshwater habitats, green algae (e.g., *Chlorella* spp.) form intracellular symbioses with a variety of heterotrophic host taxa, and these types of "nutritional mutualisms" (*Clark et al., 2017*) are essential in aquatic habitats (*Smith & Douglas, 1987*; *Reiser, 1992*).

Despite their importance, many facets of the molecular and cellular interactions that allow long-term partnerships remain obscure for a range of phototroph:heterotroph symbioses (*Hill & Hill, 2012*). To date, our understanding of freshwater:algal intracellular symbiosis has largely been informed by two *Chlorella*-based symbioses found in *Paramecium* and *Hydra* host backgrounds (e.g., *Kodama & Fujishima, 2010*; *Kovacevic, 2012*). *Hydra:Chlorella* symbioses were among the first animal systems to conclusively demonstrate the transfer of photosynthetically-fixed carbon from the symbiont to the host (*Muscatine & Hand, 1958*) and *Paramecium:Chlorella* symbioses have long been known to benefit host growth (*Karakashian, 1963*). Molecular and cellular tools have shed further light on the symbioses revealing that a highly coordinated series of cellular and molecular events transpires as *Chlorella* are taken up by *Paramecium* (*Kodama & Fujishima, 2010*), and a unique set of genes are up and down regulated in the host in response to establishment of the symbiosis in *Paramecium* with and without *Chlorella* symbionts (*Kodama et al., 2014*). Among the mechanisms that appear to be regulated during endosymbiosis, glutamate and glutamine biosynthesis has been speculated to play roles in nitrogen metabolism. For example, *He et al. (2019)* demonstrated that *Paramecium bursaria* regulate abundance of their symbionts through glutamine supply. Distinct gene expression patterns during endosymbiotic interactions between two species of *Hydra* and their algal symbionts has also been revealed (*Ishikawa et al., 2016*) and interestingly, glutamine synthesis seems to play a key role in this symbiosis as well (*Hamada et al., 2018*).

While *Chlorella*-based symbioses have been predominantly studied in both *Paramecium* and *Hydra*, photosynthetic green algal symbionts other than *Chlorella* are also found in many species, and it is clear that intracellular green algal symbioses have evolved multiple times over the course of evolution (*Hoshina & Imamura, 2008*; *Rajević et al., 2015*). An important characteristic of these symbioses is the degree of intimacy between partners, and obligacy is the pinnacle of coevolutionary specialization (e.g., *Amann et al., 1997*). However, the initial interactions involving intracellular occupancy likely involved some degree of ephemerality without tight integration between partners (*Strehlow et al., 2016*). Even for well-studied symbioses, specific factors that permit long-term residency of a symbiont within a host cell often remain obscure (*Hill, 2014*; *Clark et al., 2017*). A comparative approach is especially useful if we hope to understand the forces that shape long-term mutualistic symbioses that lead to obligacy. For example, *Bosch, Guillemin & McFall-Ngai (2019)* recently highlighted the development and use of several laboratory symbiosis model systems that will help construct a more complete picture of host-microbe interactions including several early branching animals (e.g., *Nematostella vectensis*, *Aiptasia pallida*, *Hydra vulgaris*). They argue that interrogating a variety of "evolutionary 'experiments' in
symbiosis" will shed light on the mechanisms and diversity of these interactions and lead to better understanding of how animals have evolved, making the case that future studies should include identifying mechanisms for symbiosis in sponge holobionts.

Freshwater sponges from several genera harbor green algal species and these partnerships were an early focus of study for scientists interested in symbiosis (*Brøndsted & Brøndsted, 1953*; *Brøndsted & Løvtrup, 1953*; *Muscatine, Karakashian & Karakashian, 1967*; *Gilbert & Allen, 1973a*; *Gilbert & Allen, 1973b*; *Wilkinson, 1980*). Much of the initial work centered on the ecological importance of photosynthetic sponges in freshwater ecosystems (e.g., *Williamson, 1977*; *Williamson, 1979*; *Frost & Williamson, 1980*) yet freshwater sponge symbioses are poorly represented in the modern algal-based symbiosis literature. The emergence of powerful molecular tools, however, offers renewed opportunities to study sponge-based symbiotic systems, which is aided by the fact that freshwater sponges offer many tractable qualities of a model system (*Kenny et al., 2019*; *Kenny et al., 2020*). With modern molecular and cellular tools, however, freshwater sponges are proving to be an exciting tool to study intracellular symbiosis.

We demonstrate here that the sponge *Ephydatia muelleri* is an excellent model to study symbiosis. The genus *Ephydatia* belongs to the Spongillidae, a species rich family of exclusively freshwater haplosclerid demosponges. It has a pancontinental distribution, which may be due, at least in part, to transportation in guts (*McAuley & Longcore, 1988*) or on feathers (*Manconi & Pronzato, 2016*) of foraging waterfowl. It produces diapausing cysts (i.e., gemmules) that can withstand freezing and be stored at −80 °C (*Leys, Grombacher & Hill, 2019*), and thousands of clonal individuals can be cultured at room temperature with minimal lab equipment (*Barbeau, Reiswig & Rath, 1989*). Due to the facultative nature of the sponge:symbiont partnerships, the green algal symbiont can often be easily cultured outside of the host, and, as we show here, sponges can grow with and without the algal symbionts.

Recently, a high quality *E. muelleri* genome was sequenced with chromosomal-level assembly with RNASeq data for four developmental stages (*Kenny et al., 2020*). *E. muelleri* is also amenable to a variety of cellular, genetic, and molecular approaches that allow researchers to study gene function (e.g., *Windsor & Leys, 2010*; *Rivera et al., 2011*; *Schenkelaars et al., 2016*; *Schippers & Nichols, 2018*; *Windsor Reid et al., 2018*; *Hall et al., 2019*). These aspects of sponge:algal cultivation along with the molecular resources make *E. muelleri* a promising model system to study host:symbiont integration and specialization at a cellular and genetic level to identify mechanisms that shape integration between hosts and symbionts. Here we evaluate host:symbiont interactions by examining the fate of sponge-derived *Chlorella*- like green algae introduced to aposymbioitc sponges recently hatched from gemmules. We identify putative genetic pathways involved with establishing the endosymbiosis through RNASeq analysis and we discuss the implications of this work in light of growing interest in understanding general mechanisms that may guide symbiotic interactions.

## MATERIALS AND METHODS

### Sponge and algal collection

*Ephydatia muelleri* gemmules were collected in the winter months from shallow, rocky streams at the base of dams in Richmond, VA in Bryan Park (37.598047, −77.468428) under Virginia Department of Game and Inland Fisheries Permit #047944. Gemmule-containing sponges were located on the undersides of rocks, and samples were transported on ice in foil-wrapped, 50 ml conical tubes. In the lab, gemmule-containing sponge tissue was placed in cold 1× Strekal's solution (*Strekal & McDiffett, 1974*) in a petri dish, and under a microscope illuminated with low light, gemmules were separated from residual adult skeletal material. Isolated gemmules were washed in a weak hydrogen peroxide solution (2%) before being stored at 4 °C in 1×Strekal's or in 20%DMSO at −80 °C (*Leys, Grombacher & Hill, 2019*).

Algae-bearing sponges were identified in summer months based on their bright green coloration, and sponges were returned to the lab for algal isolation. A small piece ($\approx 1$ cm$^3$) of clean tissue was removed from the sponge, and then washed multiple times in 1X Strekal's solution. Cleaned sponge tissue was then ground in 1X Bold Basal Medium (BBM; Sigma-Aldrich, Milwaukee, WI) in a clean, acid-washed mortar and pestle. Algae in the resultant slurry were allowed to precipitate and the supernatant was removed and replaced with fresh 1X BBM. This process was repeated multiple times to create an algal-enriched solution. Once nearly all visible sponge material was removed, 1 µl of the algal suspension was added to 200 ml of sterile BBM. Algal growth was obvious within 1 week. Algal cultures were subsequently plated onto BBM agar plates for the isolation of individual algal colonies. Algal lines were grown continuously in either Basal Medium (Sigma-Aldrich, Milwaukee, WI) or in Modified Bolds 3N Medium (UTEX, Austin, TX, USA).

### Algal cultures and identification

Algae were propagated at ±25 °C under fluorescent light for 16 h per day. DNA from cultured algae was isolated using the CTAB procedure, and 18S rDNA was sequenced. PCR amplification of 18s rDNA was done using protist specific molecular barcoding primers E528F, N920R, GF, GR, BR, and ITS055R (*Marin, Klingberg & Melkonian, 1998*; *Marin et al., 2003*). PCR conditions included 4 min at 94 °C; 30 cycles of 30 s at 94 °C, 30 s at 55 °C, and 45 s at 72 °C. A final elongation step of 2 min at 72 °C was included. PCR products were separated on a 1% agarose gel to verify amplification. Amplicons were cleaned using the QIAquick PCR Purification Kit (Qiagen, Hilden, Germany) and sequenced. Additional markers for identification of *Chlorella* spp. isolates for nuclear SSU and chloroplast SSU were also used (*Wu, Hseu & Lin, 2001*) and products were sequenced as described. All sequences are provided in File S1.

### Algal infection of sponges

*Ephydatia muelleri* was grown from gemmules in 1X Strekals in 6 well plates over a three to five-day period, which corresponded to the development of a mature canal system with osculum and evidence of active pumping (*Leys, Grombacher & Hill, 2019*). Live sponge-derived algal cells were introduced into the water surrounding the sponge. We

initiated all infections with 130,000 algal cells ml$^{-1}$1X Strekal's harvested during the logarithmic portion of their growth phase. We estimated cell densities and population growth characteristics using optical density (OD) measurements at 425 nm and 675 nm, which had been correlated with actual cell counts determined with a hemocytometer. Algae were slowly pipetted around and above the tissue to inoculate sponges. Infected sponges were placed under a 12:12 light:dark exposure.

## Microscopy

For confocal microscopy, sponges were grown in 35 mm glass bottom dishes (MatTek Life Sciences) and sponge tissue with and without algae was fixed in 4% paraformaldehyde and 1/4 Holtfreter's Solution overnight at 4 °C. Tissue was washed three times in 1/4 Holtfreter's Solution, permeabilized with 0.1% Triton X-100/PBS for three minutes, and washed three times in PBS. Tissue was stained with Hoescht 33342 (1:200 dilution, Thermo Fisher Scientific, Waltham, MA, USA) and Phalloidin Alexa 488 (1:40 dilution, Thermo Fisher Scientific, Waltham, MA, USA) in PBS and incubated in the dark for 20 min, washed three times in PBS and imaged imaged using an Olympus FV1200 laser scanning microscope using FluoView software.

For electron microscopy, sponge samples infected with algae were fixed in 2.5% glutaraldehyde in sterile filtered water for 1 h at room temperature and then overnight at 4 °C. Fixed samples were washed in 0.2 M cacodylate buffer (pH 7.4) and postfixed with 1% OsO4 and 1% Uranyl acetate. Samples were dehydrated in an ethanol series, infiltrated in propylene oxide, and embedded in Embed 812 plastic resin. After polymerization, one mm sections were cut and treated for 1 h in 4% hydrofluoric acid:76% ethanol to dissolve spicules. These sections were then re-dehydrated, re-infiltrated, and re-embedded following the protocol described above. Ultrathin sections were stained with uranyl acetate and quick lead. Micrographs were taken using a JEOL 1010 transmission electron microscope.

## RNA isolation, library construction, and sequencing

Sponges were grown from gemmules in 1X Strekal's to the stage where a functioning osculum had developed. To triplicate samples of these sponges (∼20–30 sponges per treatment), we added live algal cells (130,000 *Chlorella* ml$^{-1}$) or no algae as treatments. Tissue was collected after 24 h of exposure to algae, washed several times to remove algae from the surrounding water and surfaces, and either stored at −80 °C after RNA*later* treatment (Thermo Fisher Scientific, Waltham, MA, USA) or processed immediately for RNA. Total RNA was isolated using the animal tissue RNA purification kit (Norgen Biotek, Thorold, Ontario, Canada). Total RNA was sent to LC Sciences (Houston, TX, USA) where RNA integrity was checked with Agilent Technologies 2100 Bioanalyzer (Agilent, CA). Ribosomal RNA was removed at LC Sciences using Ribo-Zero ribosomal RNA reduction, followed by fragmentation with divalent cation buffers in elevated temperature. Sequencing libraries were prepared by LC Sciences following Illumina's TruSeq-stranded-total-RNA-sample preparation protocol (Illumina, San Diego, CA, USA). Quality control analysis and quantification of the sequencing library were performed using Agilent Technologies 2100 Bioanalyzer High Sensitivity DNA Chip. Paired-ended sequencing was performed on Illumina's NovaSeq 6000 sequencing system by LC Sciences.

## Transcript assembly and analysis

Cutadapt 1.10 (*Martin, 2011*) and proprietary perl scripts (LC Sciences) were used to remove the reads that contained adaptor contamination, low quality bases and undetermined bases. Sequence quality was verified using FastQC 0.10.1 (http://www.bioinformatics.babraham.ac.uk/projects/fastqc/). Two methods were used for transcript assembly. In one analysis, Bowtie 2 (*Langmead & Salzberg, 2012*) and HISAT 2.0 (*Kim, Langmead & Salzberg, 2015*) were used to map reads to the reference genome of *E. muelleri* (*Kenny et al., 2020*). The mapped reads (bam format) of each sample were assembled using StringTie (*Pertea et al., 2015*). All transcriptomes from 6 samples were merged to reconstruct a comprehensive transcriptome using perl scripts and gffcompare (https://github.com/gpertea/gffcompare/). After the final transcriptome was generated, StringTie (*Pertea et al., 2015*) and edgeR (*Robinson, McCarthy & Smyth, 2010*) were used to estimate the expression levels (FPKM) of all transcripts and genes across all replicate samples. mRNAs with log2 (fold change) >1 or log2 (fold change) <-1 and with statistical significance where the *p*-value was <0.05 were considered to be differentially expressed at a significant level. Gene Ontology (GO) and KEGG annotation and enrichment analysis of differentially expressed genes was performed. In a second analysis, de novo assembly of the transcriptome was performed with Trinity 2.4 (*Grabherr et al., 2011*). Quality of the assembled result was judged by length of unigenes, GC content, and N50. All assembled Unigenes (longest transcripts in clusters of 'genes' based on shared sequence content) were aligned against the non-redundant (Nr) protein database, GO, SwissProt, KEGG and eggNOG databases using DIAMOND (*Buchfink, Xie & Huson, 2014*) with a threshold of Evalue<0.00001. Salmon (*Patro et al., 2017*) was used to perform expression level for unigenes by calculating TPM (*Mortazavi et al., 2008*). The differentially expressed unigenes were selected with log2 (fold change) >1 or log2 (fold change) <-1 and with statistical significance (*p* value <0.05) by R package edgeR (*Robinson, McCarthy & Smyth, 2010*).

## RESULTS

### Algal symbionts can be cultivated outside of freshwater sponge hosts

Freshwater sponges from the field are observed with and without symbionts, even within the same individual, depending on growth locations and exposure to light (Fig. 1). Symbiotic algae were isolated from *Ephydatia muelleri*, cultured, and DNA sequencing indicated that the isolate belongs to the Chlorelleaceae (File S1). The strain is *Chlorella*-like in morphology, grows well in commercially available algal media across a range of temperatures (16 °C to 25 °C) and light:dark regimes (12:12, 16:8, 24:0). Due to its easily culturable nature, we have continuously grown this strain for more than five years in the lab. Our *Chlorella*-like isolate reached a stationary phase of growth (approximately $1.0 \times 10^8$ cells/ml) by 15 days when grown under the standard conditions used for growing freshwater sponges in the lab (22−23 °C, 16:8 light:dark). The algae also grew well on BBM plates and individual colonies were used to make frozen stocks of the algal strain.
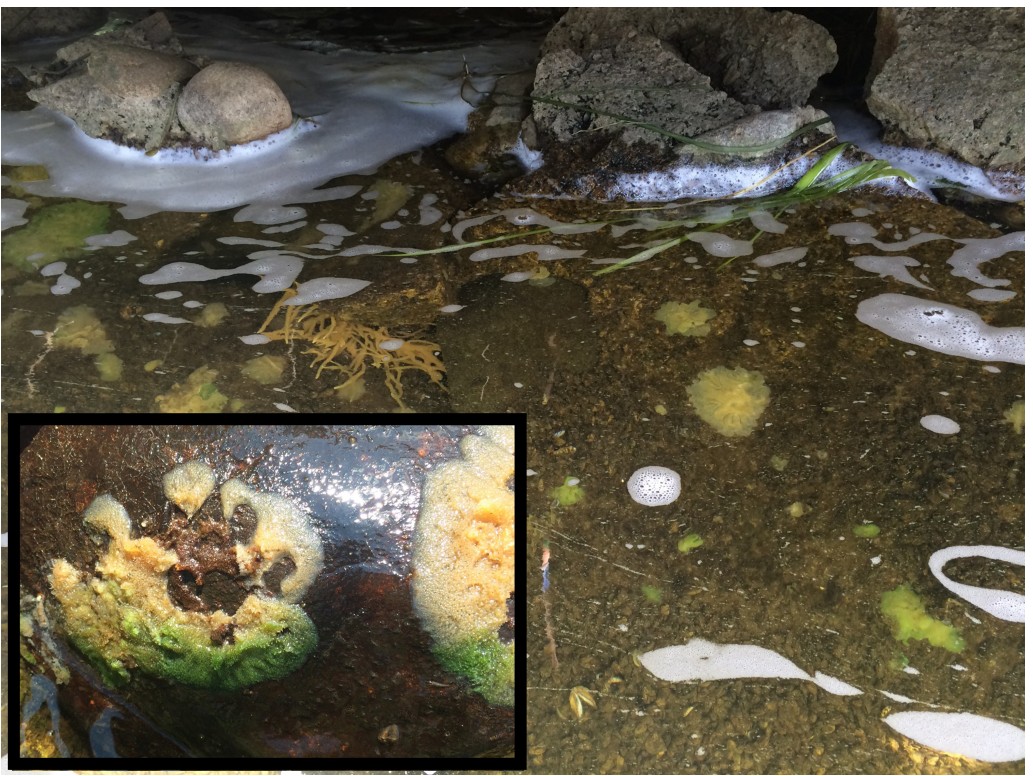

**Figure 1 Freshwater sponges in natural habitat growing at the outflow of the dam.** Several sponge species are present - some harbor green algae, some do not. (Inset) Example of sponge harboring green algae. The sponge was growing on the underside of a rock, which has been turned over. The portion of the sponge that would have been exposed to sunlight (bottom portion of the sponge) is green due to the presence of algal. Tissue protected from sunlight is devoid of algae (top portion of sponge colony).

## Sponge-derived algal symbionts stably infect aposymbiotic *E. muelleri*

Aposymbiotic *E. muelleri* sponges were hatched from gemmules and grown to full development at stage 5 (*Kenny et al., 2020*). At this point, sponge-derived *Chlorella*-like symbionts in exponential growth phase were added to the media. The infected sponges had extensive canal systems and functioning oscula (Fig. 2). The majority of algal cells captured by *E. muelleri* appeared to be located in intracellular compartments by 24 h post infection as observed by confocal microscopy (Fig. 3). Evidence of the establishment of intracellular residence by the algae was apparent within 4 h of infection (Fig. 4A). At the 24 h time point, however, we observed many sponge host cells that harbored single or multiple algae within a single cell (Figs. 4B & 4C; 5) and few algae that remained in extracellular locations. Persistence of algae within host cells through 6 days was obvious, though we observed that algae-containing sponge cells shifted location and were concentrated around and adjacent to choanocyte chambers (Fig. 3D).

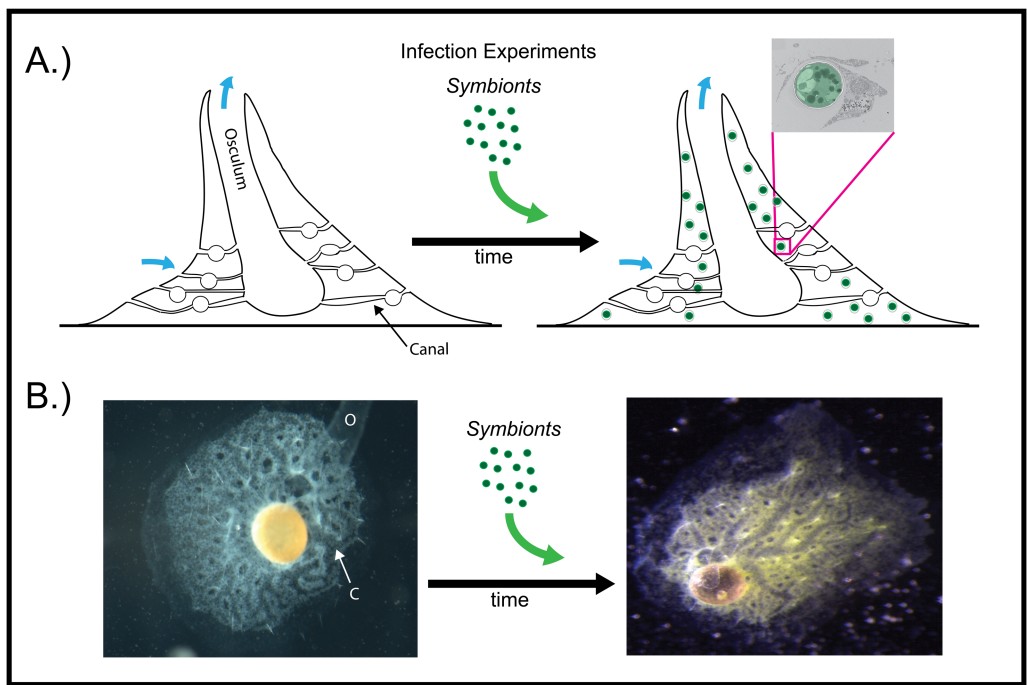

**Figure 2** **Infection of aposymbiotic *E. muelleri* sponges.** (A) Schematic of infection process. Inset shows electron micrograph of algal engulfment by sponge cell. (B) *E. muelleri* without algae and 24 h post-infection with algal symbionts. O (osculum), C (canal).

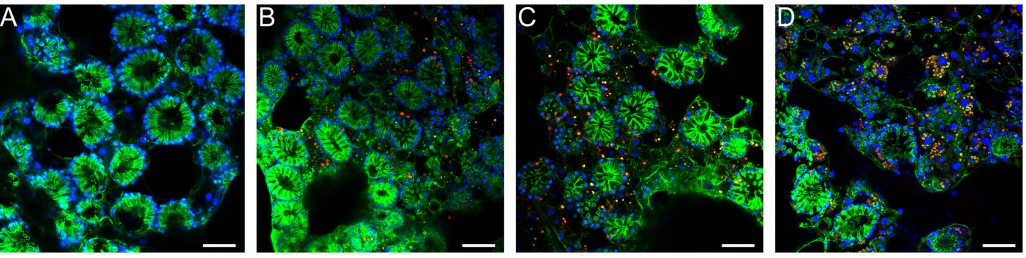

**Figure 3** **Confocal time series of *E. muelleri* choanoderm region after infection with *Chlorella*-like symbionts.** (A) Aposymbiotic *E. muelleri* (B) *E. muelleri* 4 h post-infection. (C) *E. muelleri* 24 h post-infection. (D) *E. muelleri* 6 days post-infection. Note cells with multiple algae. Images show DNA in blue, F-actin in green, and autofluorescence of algal cells in red. Scale bars 30 μm.

## RNA Sequencing, assembly, and mapping to the *E. muelleri* genome

Six cDNA libraries, three from aposymbiotic *E. muelleri* and three from *E. muelleri* 24 h post-infection with sponge-derived algae, were constructed and sequenced on the Illumina NovaSeq 6000 platform. Quality control and read statistic data for each sample are given in Table S1, with results shown before and after read cleaning. Sequencing quality was exceptionally good, with high (>98%) Q30% observed for all samples. The least well-recovered sample was EmInf3, with 7.54 Gbp sequenced, and the most-sequenced sample, EmInf1, contained 10.32 Gbp. In all cases, a good level of sequencing depth was

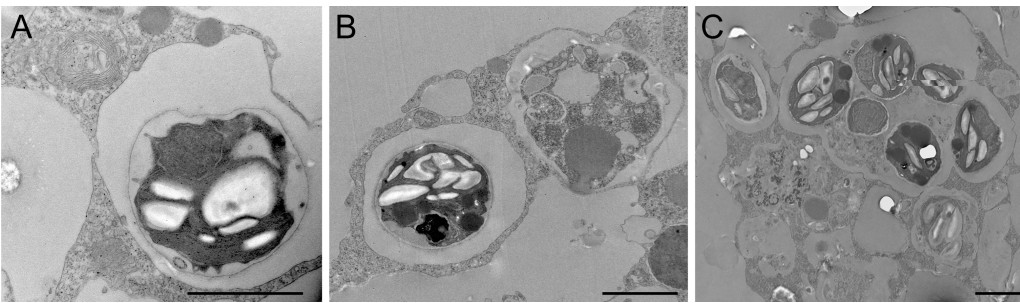

**Figure 4  Transmission electron microscopy of intracellular algal symbionts after *E. muelleri* infections.** (A) *E. muelleri* 4 h post-infection. (B) Multiple infected cells 24 h post-infection. C. Once cell with multiple algal symbionts 24 h post-infection. Scale bars 2 μm.

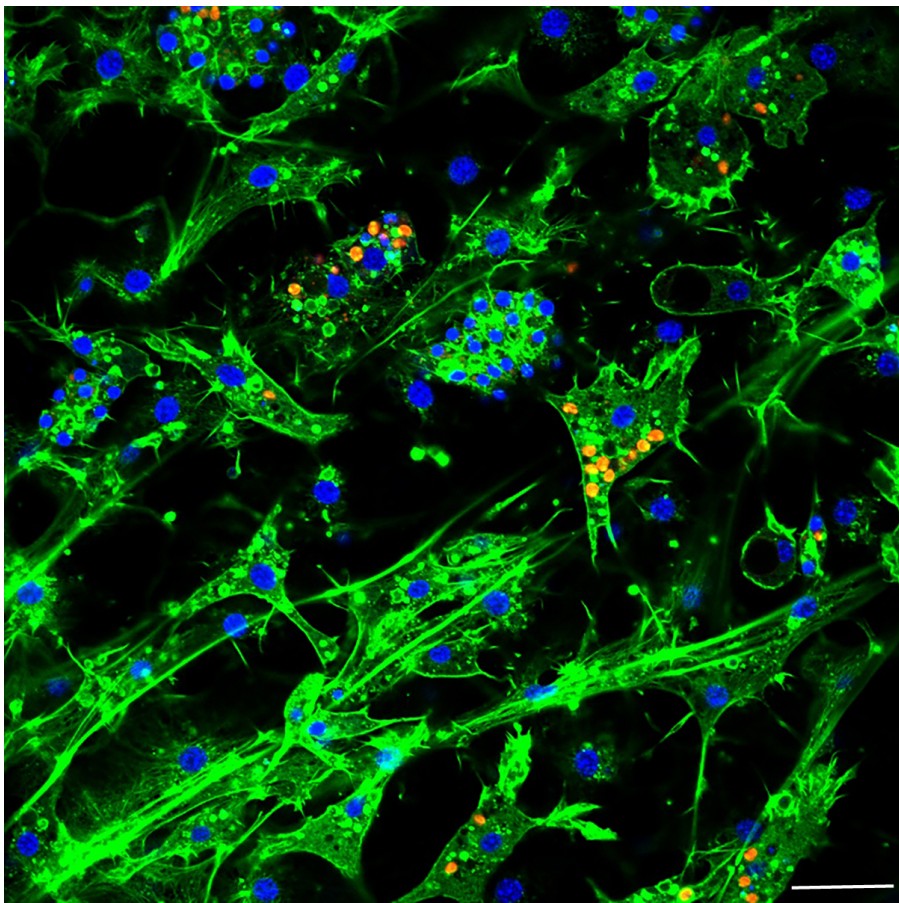

**Figure 5  Confocal image at 24 h post-infection showing multiple intracellular algal symbionts in one sponge cell.** Images show DNA in blue, F-actin in green, and autofluorescence of algal cells in red. Scale bars 20 μm.

observed for three samples per stage. A total of 65,377,412 raw reads with a $Q_{20}$ value of 99.98% were generated for the aposymbiotic sponges and 59,214,624 raw reads with a $Q_{20}$ value of 99.98% were generated for the symbiotic sponges. After removing the low-quality sequences, short reads and ambiguous nucleotides, the remaining valid reads were 63,552,928 for the aposymbiotic treatment and 57,705,518 for symbiotic treatments. For all replicate samples, good mapping results were observed to the reference genome (*Kenny et al., 2020*). In any sample, no fewer than 56.50% of all reads could be mapped to the *E. muelleri* genome and 36,771,764 (57.87%) mapped reads and 22,562,710 (35.5%) unique reads were obtained for the aposymbiotic sequences while 33,145,990 (57.36%) mapped reads and 20,718,294 (35.96%) unique reads were found for the symbiotic sequences (Table S2). The number of raw reads mapped to each *E. muelleri* gene or transcript is given in Table S3. Of the reads that map to the genome, greater than 70% of reads were placed in exonic regions for all samples and less than 1% of the RNASeq reads mapped intergenically (Fig. S1).

Normalizing of expression units was performed using FPKM for both gene and transcript expression and FPKM interval chart and density graphs comparing overall gene expression between samples (Fig. S2, Table S4) reveal that variation in expression between samples is low and distinct distributions are nearly the same for each sample. This indicates that the quality of data obtained by sequencing was reliable for further analysis. Even though we do not yet have an available reference genome for the native sponge-derived algae or for the bacterial symbionts present in our dataset, we believe that the overall transcriptome data sets, including de novo assembly of the transcriptomic data and functional annotation of unique genes expressed by the algae in the symbiotic state will be of interest to others who study symbionts or are interested in non-coding RNA as we used total RNA sequencing to capture a broader range of gene expression changes (i.e., transcripts in both coding and non-coding RNA). We also used RNA depletion rather than poly-A tail selection.

## De novo reconstruction of transcriptomes from RNA-Seq data

In order to elucidate genes expressed in the native algae during endosymbiosis, we also report a de novo assembly and functional annotation of the transcriptomic data set. While the assembly and RNA-Seq analysis described above compared expression profiles of sponge genes during apopsymbiotic and symbiotic states, the de novo assembly also reveals a set of algal transcripts expressed during the symbiosis. In all, there were 106,175 total predicted transcripts with a minimum length of 201 bp and maximum of 40,322 bp (median length 666 bp) from the de novo assembly. The GC content was 47.97% with an N50 of 1,605. Predicted genes, including sponge and algal, were calculated at a total of 22,914 with a GC content of 48.11% (median length 573 bp) and N50 of 1,715. We attempted to map the transcriptome data to some published Chlorella genomes (e.g., *C. sorokiniana*, *Chlorella sp*. A99), but found that low mapping rates prohibited alignment against these reference genomes. Thus, the *Chlorella*-like native symbiont described here belongs to a different lineage and it will be necessary to sequence the genome of this strain in the future.

## Symbiosis-related E. muelleri genes revealed by RNASeq

To understand the genetic regulation of symbiont acquisition and maintenance from the host perspective, we examined differential gene expression at 24 h post-infection between sponges grown without algal symbionts and those that were infected with sponge-derived *Chlorella*-like symbionts. Analysis of gene expression profiles demonstrated 429 sponge genes were significantly altered (log2 >1; $p < 0.05$) between aposymbiotic and symbiotic sponges, of which 194 genes were upregulated during symbiont acquisition and 235 were downregulated (Fig. 6, File S2, Fig. S3). Transcript expression profiles demonstrated a similar pattern (Fig. S4). Among the genes with increased expression in symbiont infected sponges, 39% were either novel transcripts of unknown function or containing sequences or domains found in other organisms, but otherwise uncharacterized proteins. The genes with increased expression in aposymbiotic sponges that represent novel or uncharacterized proteins represented 46% of the dataset.

Among the enriched Gene Ontology (GO) categories revealed by the analysis, we found biological process categories to be enriched for those related to DNA catabolic processes and oxidation–reduction processes. Within the cellular component category, cytoplasm, nucleus, and membrane components were enriched. The molecular function categories included deoxyribonuclease activity, ATP binding, and metal ion binding (Fig. S5). GO enrichment analysis revealed several processes including monooxygenase activity and related oxidoreductase activity. Chitin related activities, scavenger receptor activity, receptor mediated endocytosis, DNA catabolic process, deoxyribonucleic acid activity, and multiple aspects of copper ion binding, import, and export were also enriched (Fig. 7). Using KEGG, we identified a variety of enriched pathways, including arachidonic acid, glutathione metabolism, and metabolism of molecules by cytochrome p450. Immune related signaling pathways enriched in KEGG analysis included IL-17 signaling, RIG-I-like receptor signaling, TNF signaling and NOD-like receptor signaling (Fig. 7, File S3).

The heatmap revealed changes in gene expression between infected and non-infected sponges (Fig. 6). We found that multiple loci of DBH-like and cytochrome P450-like monooxygenases, glutathione S-transferases, copper transporting ATPases, and alcohol dehydrogenases were among those upregulated in sponges infected with algal symbionts. Other noteworthy loci with increased expression in symbiotic infected sponges include leukotrienes, cholesterol 24-hydroxylase, L-amino-acid oxidase, sodium/potassium ATPase, and nmrA-like family domain-containing protein 1. Genes involved in lysosomes/phagosomes, endocytosis, or autophagy (e.g., tartrate-resistant acid phosphatase type 5-like, cathepsin L, deleted in malignant brain tumors 1) were among those increased in expression during uptake of symbionts. Genes involved in sugar metabolism (e.g., protein phosphatase 1 regulatory subunit 3B-B-like, chitin synthase 3) and signal transduction/gene regulation (e.g., transcriptional regulator Myc-A-like, cycloartenol-C-24-methyltransferase 1-like) were also represented among the genes with increased expression in the symbiotic state.

While the majority of genes with decreased expression in symbiotic sponges are present at one locus, ATP synthases, mucolipins, and E3 ubiquitin protein ligases occupy multiple loci. These genes are known to be involved in ion transport and ubiquitination as

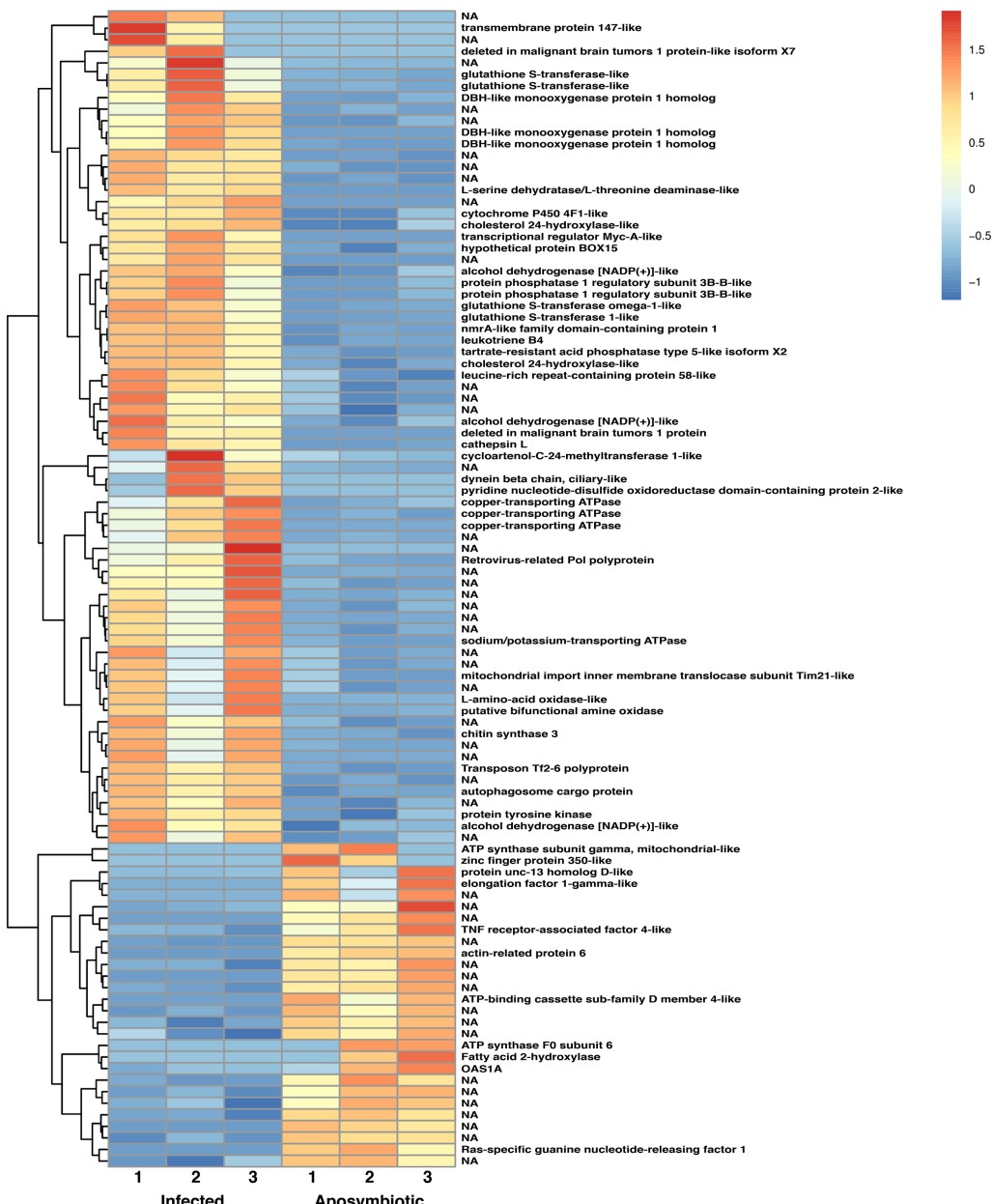

**Figure 6** **Heatmap of differentially expressed genes in RNASeq analysis.** Relative expression of differentially expressed sponge genes where red hues represent comparatively upregulated genes and blue hues represent comparatively downregulated genes (scale at right). Data shown compares triplicate samples for aposymbiotic and 24 h post-infected sponges. Gene IDs are provided at the right of each expression profile.

well as other processes. Genes involved in signal transduction or gene regulation (e.g., OAS1A, Ras-specific guanine nucleotide-releasing factor 1, kielin/chordin-like protein, serine/threonine/tyrosine-interacting-like protein 1, serine/threonine-protein kinase NIM1, NFX1-type zinc finger-containing protein 1) were often among those with lower

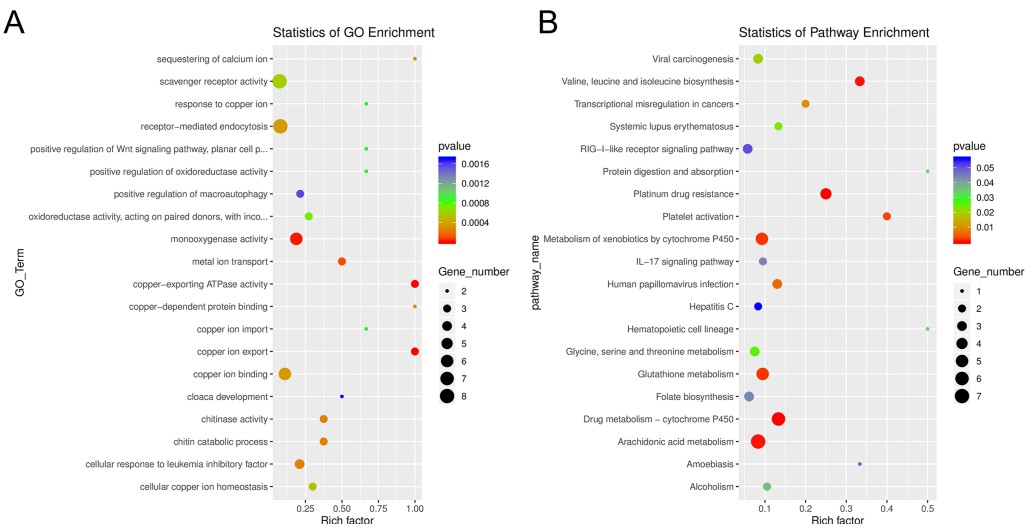

**Figure 7** **Enrichment categories for Gene Ontology and KEGG.** (A). Statistics of gene ontology (GO) enrichment across the most represented GO categories for expressed sponge genes comparing aposymbiotic and 24 h post-infection *E muelleri*. (B) Statistics of KEGG pathway enrichment across the differential sponge gene expression for aposymbiotic and 24 h post-infection *E muelleri*. Size of dots correspond to gene number while colors correspond to *p* values. Scales are given on the right.

expression in symbiotic sponges. We also found genes involved in lysosomes/phagosomes (e.g., V-type proton ATPase, ceroid-lipofuscinosis neuronal protein 6, N-acylethanolamine acid amidase) among the genes that are downregulated during symbiosis.

A few gene types had members that were either increased or decreased in response to infection by native symbionts. Four distinct glutathione S transferase genes on three different chromosomes (1, 9, and 12) showed two- to three-fold level increases in expression in symbiotic sponges, whereas expression in another glutathione S transferase (located on chromosome 5) was decreased by 1.5 fold in symbiotic sponges. A complete lack of expression in symbiotic tissue was observed for an elongation factor 1-gamma-like gene containing a glutathione S transferase domain. Sponges infected with symbionts also had increased expression for two loci of the TNF receptor-associated factor 3-like gene (both loci are clustered closely on chromosome 8). Other genes that may be involved in NF-kB signaling were also upregulated including sequestosome-1, a protein containing a Tumor Necrosis Factor Receptor (TNFR)-Associated Factor (TRAF) domain, and predicted cell death-inducing p53-target protein 1 that plays roles in regulating TNF-alpha-mediated apoptosis. Expression of a TNF receptor-associated factor 4-like gene (located on chromosome 1) and an uncharacterized protein predicted to be involved in TNF signaling and apoptosis were decreased in symbiotic sponges.

## Algal symbiont genes expressed in symbiotic state

To evaluate gene expression in the algal symbionts, blast assembled unigenes from all transcriptome treatment groups were mapped against six protein databases and statistics of annotated unigenes showed that of the 22,914 predicted genes, 34.32% could be categorized

by Gene Ontology (GO), 19.72% by KEGG, 35.26% by Pfam, 31.69% by SwissProt, 43.89% by eggNOG, and 26.43% by NR (Files S4-S9). KEGG pathway classification of the de novo assembled transcriptome which includes previously identified sponge and newly identified algal expressed genes, reveals a high percentage of algal genes involved in carbohydrate, lipid, amino acid and energy metabolism as well as genes involved in transcription, translation and protein folding, sorting and degradation, signal transduction and transport/catabolism (Fig. S6). Interestingly, functional categories predicted by eggNOG indicates the majority of genes to have unknown function, but posttranslational modification, protein turnover, and chaperones as well as signal transduction mechanisms, and intracellular trafficking, secretion, and vesicular transport all have a high number of predicted gene products (Fig. S7).

Given that aposymbiotic sponges are not expressing large numbers of algal genes, it is not surprising that profiling of differential gene expression shows many more genes upregulated in symbiotic (sponge and algal genes are both represented) compared to aposymbiotic sponges (Fig. S8). Though it is not possible to study differential gene expression in the algae by this experimental design, we did ask if there was evidence in the transcriptome data that algae were expressing genes that might defend against digestion, be involved in sugar production or transport that supply the host with photosynthetic products, or be involved in modification of the symbiosome. New categories revealed by GO enrichment that were not present in the "sponge gene only" RNASeq analysis include long-chain fatty acid biosynthetic process, cholesterol catabolic process, glucose transmembrane transporter activity, glucose import, and carbohydrate:proton symporter activity (Fig. S9). KEGG pathway enrichment reveals new categories of photosynthesis, MAPK signaling (plants), carbon fixation photosynthetic organisms, endocytosis, steroid biosynthesis, as well as a variety of small molecule metabolism (Fig. S10). For example, we find cathepsin-B and Z-like genes expressed in algae that could play roles in cellular invasion (*Que et al., 2002*), and sugar transporters (i.e., ERD6-like, bidirectional sugar transporter SWEET5) that may be involved in feeding the host (*Maor-Landaw, vanOppen & McFadden, 2019*).

All transcriptome sequences and other resources (i.e., *E. muelleri* mapped transcriptome, de novo transcriptome, gene annotations, and differential expression analysis data) are available at https://spaces.facsci.ualberta.ca/ephybase/ (under "Resources" as *E. muelleri* algal symbiosis transcriptomes; https://doi.org/10.7939/r3-7jk2-ph04). Raw data is available at NCBI Sequence Read Archive (SRA) under BioProject PRJNA656560.

## DISCUSSION

### Freshwater sponge:algal symbiosis as tractable model

As articulated by *Bosch, Guillemin & McFall-Ngai (2019)*, the use of several laboratory symbiosis model systems ("evolutionary 'experiments' in symbiosis") will help construct a more complete picture of viable pathways towards stable intracellular residency and thus animal evolution. We believe that *E. muelleri* is an excellent candidate to be a model system for these types of studies. Its ubiquity around the globe and ease of collection make it widely

available. The fact that the mutualism is facultative, with the ability to culture the organisms separately and conduct reinfection experiments (Figs. 1 and 2), offers opportunities to study pathways that permit long-term, stable residency within host cells. We have shown here that the symbiotic algae can be tracked in sponge tissues via confocal (Figs. 3 and 4) and electron microscopy (Fig. 4).

While marine sponges are important models of animal-microbe symbioses, both because they produce pharmaceutically important bioactive compounds and due to their potential to illuminate conserved mechanisms of host-microbe interactions in the basal metazoa (reviewed in *Pita, Fraune & Hentschel, 2016*), freshwater sponges should be considered as models to understand possible convergent pathways leading to intra- and extracellular symbioses. Freshwater sponges also have the added benefit of having many adaptations to freshwater systems (e.g., extreme thermal tolerance, resilience in anoxic conditions, resistance to many pollutants, ability to withstand desiccation, osmotic regulation). Recent work by *Kenny et al. (2019)* has already shown that freshwater sponges have extensive gene duplications driving evolutionary novelty and have benefited from symbioses that allow them to live in challenging conditions. Given that *E. muelleri* has a higher gene content than most animals, nearly twice that of humans (*Kenny et al., 2020*), it may not be surprising to find a large number of taxonomic specific genes among those that are differentially expressed. However, it has been noted by others that taxonomically restricted genes (TRGs) could be key to the development of species-specific adaptive processes like endosymbiosis (*Khalturin et al., 2009*; *Hamada et al., 2018*) and thus, these genes may be important in initiating or maintaining the symbioses in these sponges. Our work to adapt *E. muelleri* as a model to forward these goals should impact our future understanding of these important animals as well as the evolutionary mechanisms that shape endosymbiosis. We focus in the following sections on some of the key findings.

## Role of oxidation reduction systems in symbiotic relationships

It is well documented that oxidative environments play key roles in regulating symbiotic associations, and the interplay between regulators of redox biology have likely shaped the evolution of symbioses across life forms (*Moné, Monnin & Kremer, 2014*). Molecules involved in redox homeostasis can mediate molecular communication between hosts and symbionts as well as play roles in responses to toxic states with important pleiotropic roles for reactive oxygen and nitrogen species during the establishment of symbioses. These roles include modulation of cell division and differentiation, cellular signaling (e.g., NF-kappa B), kinase and phosphatase activities, ion homeostasis ($Ca^{2+}$, $Fe^{2+}$), and apoptosis/autophagy (*Mon, Monnin & Kremer, 2014*). Recent work in *Hydra-Chlorella* models demonstrate that symbiosis-regulated genes often include those involved in oxidative stress response (*Ishikawa et al., 2016*; *Hamada et al., 2018*). Comparisons of gene expression in *Paramecium bursaria* with and without *Chlorella variabilis* show significant enrichment of gene ontology terms for oxidation–reduction processes and oxidoreductase activity as the top GO categories (*Kodama et al., 2014*).

Given that endosymbionts are known to create reactive oxygen species (ROS) that can lead to cellular, protein, and nucleic acid damage (*Marchi et al., 2012*) and that other

symbiotic models have highlighted the importance for the host in dealing with reactive oxygen and reactive nitrogen species (RONS) (e.g., *Richier et al., 2005*; *Lesser, 2006*; *Weis, 2008*; *Dunn et al., 2012*; *Roth, 2014*; *Mon, Monnin & Kremer, 2014*; *Hamada et al., 2018*), it is not surprising that oxidative reduction system genes are differentially regulated during symbiosis in these model systems. For example, *Ishikawa et al. (2016)* show that while many genes involved in the mitochondrial respiratory chain are downregulated in symbiotic *Hydra viridissima*, other genes involved in oxidative stress (e.g., cadherin, caspase, polycystin) are upregulated. Metalloproteinases and peroxidases show both upregulation and downregulation in the *Hydra* symbiosis, and *Ishikawa et al. (2016)* show that some of the same gene categories that are upregulated in *H. viridissima* (i.e., peroxidase, polycystin, cadherin) exhibit more downregulation in *H. vulgaris*, which is a more recently established endosymbiosis. *Hamada et al. (2018)* also found complicated patterns of upregulation and downregulation in oxidative stress related genes in *Hydra* symbioses. They found that contigs encoding metalloproteinases were differentially expressed in symbiotic versus aposymbiotic *H. viridissima.*

We identified a strong indication for the role of oxidative-reduction systems when *E. muelleri* is infected with *Chlorella* symbionts (Figs. 6 and 7). While our RNASeq dataset comparing aposymbiotic with symbiotic *E. muelleri* also show differentially expressed cadherins, caspases, peroxidases, methionine-r-sulfoxide reductase/selenoprotein, and metalloproteinases, the expression differences for this suite of genes was not typically statistically significant at the 24 h post-infection time point (File S2). We find two contigs with zinc metalloproteinase-disintegrin-like genes and one uncharacterized protein that contains a caspase domain (cysteine-dependent aspartate-directed protease family) that are upregulated at a statistically significant level as well as one mitochondrial-like peroxiredoxin that is down regulated. Thus, like in the *Hydra:Chlorella* system, a caspase gene is upregulated and a peroxidase is downregulated. However, some of the differentially regulated genes we found that are presumed to be involved in oxidation reduction systems are different than those highlighted in the *Hydra:Chlorella* symbiosis. Multiple contigs containing DBH-like monooxygenases and cytochrome p450 4F1-like genes were increased in expression in symbiotic states in *E. muelleri*. Most of these genes are known to be involved in cellular oxidation–reduction systems that maintain homeostasis or act in detoxification. Oxidative stress responses have been noted in other hosts with photosynthesizing algal symbionts and may be used to deal with the reactive oxygen species (ROS) produced during photosynthesis (e.g., *Richier et al., 2005*; *Lesser, 2006*; *Hamada et al., 2018*). Interestingly, in *Aiptasia* colonized with an opportunistic *Durusdinium trenchii* compared to the same corals colonized by their native symbionts, *Breviolum minutum,* upregulation of two cytochrome P450 monooxygenases was found as well as a higher abundance of arachidonic acid (*Matthews et al., 2017*); taxonomy after (*LaJeunesse et al., 2018*). The authors speculate that this difference in lipid signaling is a result of an oxidative stress response to the non-native symbiont, but the specific role for these molecules in this system remains unclear. We do not see the wholesale upregulation of monooxygenases, as we also find that a flavin-containing monooxygenase is downregulated in the symbiotic state.

We find four loci containing distinct glutathione S transferase (GST) genes to be upregulated in *E. muelleri* infected with green algal symbionts, and one loci containing a GST gene to be downregulated during symbiosis. Interestingly, we have also noted upregulation of a GST in the marine sponge *C. varians* infected with native *Gerakladium spongiolum* (manuscript in prep). Our observation of upregulation of some GSTs and downregulation of other GSTs in sponges is enigmatic given that others seem to have found these genes to be mostly downregulated during symbiosis. *Hamada et al. (2018)* show that a GST gene is downregulated in the *H. viridissima*:*Chlorella* symbiosis. A GST was also downregulated in the symbiotic sea anemone *A. viridis* (*Ganot et al., 2011*) and in the coral *A. digitifera* infected with a competent strain of Symbiodiniaceae (*Mohamed et al., 2016*). *Kodama et al. (2014)* showed that multiple GST genes are downregulated in *P. bursaria* with *Chlorella* symbionts as compared to the symbiont free *Paramecium*. Based on observed cytological phenomena, *Kodama et al. (2014)* suggest these proteins are involved in the maintenance of the symbiosis given that the presence of algal symbionts minimizes photo-oxidative stress.

Regardless of the precise role for regulation of GSTs during endosymbiosis, the connection between glutamine supply and synthesis in both the *Paramecium* (*He et al. (2019)* and *Hydra* (*Ishikawa et al., 2016*; *Hamada et al., 2018*) systems may be an important connection. While *Hydra* most likely turn on glutamine synthetase for *Chlorella* to import nitrogen (*Hamada et al., 2018*), glutamine may also be used by the animal for synthesis and excretion of glutathione in cell growth and viability promotion or for ameliorating potential oxidative stress (*Amores-Sánchez & Medina, 1999*). Furthermore, while GSTs are best known for their role as detoxification enzymes, they are known to carry out a variety of other functions including peroxidase and isomerase activities, inhibition of Jun N-terminal kinase, binding to a range of ligands, and several novel classes of non-mammalian GSTs have functions that are not related to oxidative stress. Given the extensive gene duplication in freshwater sponges that has been described (see *Kenny et al., 2019*; *Kenny et al., 2020*) it seems possible that some of the duplicated GST genes have retained functional overlap as evidenced by their co-regulation during symbiosis, but others may have diverged to gain different functions. Investigating the role of GSTs in symbiosis regulation and dysregulation is important for uncovering new facets of host-symbiont interactions.

## Pattern recognition, innate immunity, and apoptosis

Inter-partner recognition is a key component of stable symbiotic partnerships, and host innate immunity likely plays a role in determining which microbes are targeted for destruction and which avoid detection (*Weis, 2019*). The *E. muelleri* genome possesses a variety of innate immunity genes and the upregulation of these genes occurs at stage 5 of development when the sponges have a fully organized body with ostia, canals, chambers and osculum giving them an ability to interact with the outside environment (*Kenny et al., 2020*). Given that innate immunity has been shown to play a role in coral–dinoflagellate symbiosis and the holobiont (reviewed in *Weis, 2019*) as well as in *Hydra*:*Chlorella* symbiosis (*Hamada et al., 2018*), we hypothesized that innate immune genes would be among those differentially regulated during the early stages of symbiosis.

It is well known from cnidarian-algal symbioses that microbe-associated molecular pattern (MAMP)-pattern recognition receptor (PRR) interactions are key signals playing roles in symbiont recognition and possibly maintenance of the association (reviewed in *Davy, Allemand & Weis, 2012*). We found at least one gene involved in PRR signaling pathways (i.e., deleted in malignant brain tumors 1 protein-like; dmbt1) to be expressed in symbiotic tissue, with no expression in aposymbiotic sponges. Another dmbt1-like gene containing several scavenger receptor cysteine-rich (SRCR) domains was decreased in expression in infected tissue. In addition to dmbt1-like genes, we find several other genes that may have associated scavenger receptor activity to be differentially expressed in aposymbiotic compared to symbiotic *E. muelleri*, including a tolloid-like protein (dorsal-ventral patterning tolloid-like protein 1) and several sponge-specific uncharacterized proteins (Em0017g780a, Em0083g1a, Em0017g784a, Em0742g1a - all of which were downregulated). It is possible that these PRRs play an important role in freshwater sponge-green algal recognition. Dmbt1 is a multiple SRCR domain containing glycoprotein implicated in immune defense and epithelial differentiation (*Mollenhauer et al., 2000*). Scavenger receptors are a class of PRRs that may function in recognition and regulation in cnidarian–Symbiodiniaceae symbioses (*Weis, 2019*). We previously showed that dmbt1 exhibited increased expression in aposymbiotic *Cliona varians* compared to *C. varians* infected with its *G. spongiolum* symbiont (*Riesgo et al., 2014*). Dmbt1 is downregulated upon bacterial challenge in oysters (*McDowell et al., 2014*) and the coral *Acropora millepora* (*Wright et al., 2017*). In the case of *A. millepora*, it was suggested that dmbt1 may play a role in maintaining symbiotic associations with commensal microbes. In addition to SRCR domains, this dmbt1 gene also contains a calcium-binding EGF-like domain characteristic of membrane-bound proteins that require calcium binding for protein-protein interactions.

Other molecules may also play a role in pattern recognition. For example, we observed decreased expression of two different sushi, von Willebrand factor type A genes. These types of complement control domain containing proteins (CCP) are often involved as pattern recognition molecules in determining "self" vs. "non-self." The multiple CCP we found have receptor–ligand interaction regions, and their downregulation suggests potential influence of the symbiont on host expression patterns. As regulators of complement activation, CCPs can protect cells by interacting with components of the complement system or through activation of immune cells and processing of immune complexes when dealing with microbes and other foreign materials (*Hourcade, Holers & Atkinson, 1989*).

We also identified 15 differentially regulated contigs included in the KEGG enrichment data set that were involved in the nucleotide-binding oligomerization domain-like receptor (NLR) signaling pathway. These NLR are important components of innate immunity involved in cytoplasmic recognition of pathogen- and damage-associated molecular patterns (PAMPs and DAMPs, respectively) that specifically recognize "non-self" components of the cell (*Creagh & O'Neill, 2006*). The NLR signaling pathway initiates signaling cascades that lead to regulation of NF-kB and MAPK pathways. One of the genes associated with NOD-like receptor signaling is Oas1a, which was downregulated in our symbiotic sponges. Oas1a is an interferon-induced, dsRNA-activated antiviral enzyme that plays roles in innate immunity and apoptosis. In addition to the typical

2′–5′-oligoadenylate synthetase 1 and Nucleotidyltransferase (NT) domains, the Oas1-like gene that we found contains a TPR repeat (signal transduction) domain as well as three MYND finger domains, a probable pectinesterase domain, and two parallel beta helix regions that share some similarity with pectate lyases. Whether pectin-moieties on the surface of the symbiont are a target, and thus involved in symbiont acquisition, remains to be seen. Three contigs related to MAPK signaling were also differentially regulated, including the Ras-specific guanine nucleotide-releasing factor 1 which was decreased in expression in symbiotic *E. muelleri*. Further experiments will be needed to ascertain how these pathways are involved in initial uptake or maintenance of the symbiosis.

We found differentially expressed contigs related to innate immunity and apoptosis functions. In particular, upregulation of two TNF receptor-associated factor 3-like genes and downregulation of one TNF receptor-associated factor 4-like gene suggests a role for immune function or apoptosis. TNF receptor-associated factor 4-like genes regulate activation of NF-kappa-B in response to signaling through Toll-like receptors whereas TNF receptor-associated factor 3-like genes tend to act as negative regulators of NF-kappa-B activity; both are involved in apoptotic processes. We observed (1) upregulation of a tartrate-resistant acid phosphatase type 5-like gene in symbiotic tissue, which has GO categorization of negative regulation of tumor necrosis factor (TNF) production; (2) upregulation of cell death-inducing p53-target protein 1, which is known to regulate TNF-alpha-mediated apoptosis; and (3) upregulation of sequestosome-1, an autophagosome cargo protein that is also known to regulate TNF receptor associated factors as well as NF-kappa-B in some cellular contexts (*Kim & Ozato, 2009*). In addition to these genes, we found other contigs with transcripts predicted to be involved in Toll-like receptor/NF-kappa-B/TNF-receptor signaling and apoptosis amongst the sponge-specific uncharacterized and/or predicted proteins that are differentially regulated in symbiotic states (File S2; Em0002g1214a, Em0023g342a, Em0084g5a). The coral-Symbiodiniaceae literature provides evidence that symbionts may be modulating the host immune response via repression of NF-kappa-B (e.g., *Weis, 2019*), and while more work will need to be done to determine if NF-kappa-B function is repressed, our data suggests the involvement of the TNF pathway in modulating the symbiosis.

## Nitrogen metabolism

Nitrogen has long been suspected to be a key factor in the regulation of symbiont populations in hosts (*Radecker et al., 2015*), though regulatory connections between host and symbiont are generally poorly understood. For photosynthetic symbionts, nitrogen demands are elevated due to the photosynthetic apparatus, and nitrogen metabolism is a key feature of digestive processes of heterotrophic hosts. Thus, there seem to be opportunities for host:symbiont coevolutionary specialization in terms of nitrogen metabolic integration.

In the *Hydra*:*Chlorella* symbiosis, glutamine synthetase (GS-1) expression was found to be elevated in host tissue when *Chlorella* symbionts were present and when the host was exposed to maltose (*Hamada et al., 2018*). Indeed, GS-1 was one of the four main genes shown to be specifically upregulated in *H. viridissima* by the presence of *Chlorella* symbionts. *Hamada et al. (2018)* demonstrated that the symbiotic *Chlorella* could not use

nitrite and ammonium as nitrogen sources, and instead relied upon *Hydra* for nitrogen assimilation through the action of glutamine synthetase and the uptake and processing of ammonium to glutamine. While we do not find glutamine synthetase to be upregulated in *E. muelleri* (at least not at 24 hr post-infection), we do find an asparagine synthetase (File S2) to be significantly increased in expression in symbiotic compared to aposymbiotic sponges. Asparagine is a major nitrogen transporter in plants and asparagine synthetase, using glutamine as a substrate, is a key enzyme involved in the regulation of carbon-nitrogen balance in plants through nitrogen assimilation and distribution (e.g., *Qu et al., 2019*). Thus, upregulation of asparagine synthetase here may indicate that the algae are using similar processes for nitrogen regulation. Future experiments aimed at analysis of growth parameters for this symbiotic strain of green algae using different nitrogen and sugar sources could help increase our understanding of nitrogen metabolism in this regard.

Two primary models have been proposed to explain hypothesized use and uptake of nitrogen in symbioses involving heterotrophic hosts and phototrophic symbionts (see *Wang & Douglas, 1998*). The first is the straightforward hypothesis that symbionts assimilate nitrogenous waste (primarily ammonium) from the host and translocate it back to the host in other forms. The second is the more complicated hypothesis that symbiont-derived carbon compounds reduce host catabolism of nitrogenous compounds. Our data do not permit favoring one of these hypotheses, but the potential regulation of a key enzyme in nitrogenous pathways deserves greater attention given the importance of this element to photosynthetic efficiency and as a vehicle for host:symbiont integration.

While the goal of this work was to demonstrate the utility of *E. muelleri* as a model system for studying endosymbiosis with algae, and to study host differential gene expression in response to algal symbionts, we also report a set of genes from de novo transcriptome assembly that are expressed in the green algae when they are endosymbiotic. Future work directed at sequencing the native symbiont genome, as well as comparisons of the symbionts in their cultured, free-living form versus those isolated from the host intracellular environment will be essential to understanding the molecular regulatory mechanisms adopted by the algae *in hospite* compared to the free-living environment and will hopefully provide more clues about the pathways utilized by both host and algae in establishing and maintaining this symbiosis.

## CONCLUSIONS

We demonstrate the utility of a *E. muelleri*:chlorophyte symbiosis to identify features of host cellular and genetic responses to the presence of intracellular algal partners. Freshwater sponges and their symbiotic partners are easy to maintain under laboratory conditions, and the genomic and transcriptomic data available for the host offer powerful experimental opportunities. The freshwater sponge system also offers an important comparative perspective when placed in the context of work done with *Paramecium*, *Hydra*, and other heterotrophic:phototrophic symbioses. Our work demonstrates that freshwater sponges offer many tractable qualities to study features of intracellular occupancy and thus meet many criteria desired for a model system.

## ACKNOWLEDGEMENTS

We are thankful to Omar Quintero for assistance with confocal microscopy and Sally Leys for hosting the transcriptomic resources on EphyBase.

### Funding

This work was supported by funding from the National Science Foundation (Award # 1555440) to April L. Hill and Malcolm S. Hill and by an Institutional Development Award (IDeA) from the National Institute of General Medical Sciences of the National Institutes of Health under grant number P20GM103423. The funders had no role in study design, data collection and analysis, decision to publish, or preparation of the manuscript.

### Grant Disclosures

The following grant information was disclosed by the authors:
National Science Foundation: #1555440.
Institutional Development Award.
National Institute of General Medical Sciences of the National Institutes of Health: P20GM103423.

### Competing Interests

The authors declare there are no competing interests.

### Author Contributions

- Chelsea Hall, Malcolm S. Hill and April L. Hill conceived and designed the experiments, performed the experiments, analyzed the data, prepared figures and/or tables, authored or reviewed drafts of the paper, and approved the final draft.
- Sara Camilli performed the experiments, analyzed the data, prepared figures and/or tables, authored or reviewed drafts of the paper, and approved the final draft.
- Henry Dwaah, Benjamin Kornegay and Christie Lacy performed the experiments, prepared figures and/or tables, and approved the final draft.

### Field Study Permissions

The following information was supplied relating to field study approvals (i.e., approving body and any reference numbers):

Field collection was conducted under Virginia Department of Game and Inland Fisheries Permit #047944.

### Data Availability

The transcriptome sequences and gene annotations are accessible at the EphyBase:

Hill AL, Camilli S, Dwaah H, Kornegay B, Lay CA, Hill MS (2020). ''Freshwater sponge hosts and their green algae symbionts: a tractable model to understand intracellular

symbiosis". Education & Research Archive (ERA). Dataset. https://doi.org/10.7939/r3-7jk2-ph04.

Raw sequence files are available at NCBI SRA under BioProject PRJNA656560: accession numbers: SRS7182819, SRS7182818, SRS7182817, SRS7182816, SRS7182815, SRS7182814.

## Supplemental Information

Supplemental information for this article can be found online at http://dx.doi.org/10.7717/peerj.10654#supplemental-information.

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
