# Peer review of "Freshwater sponge hosts and their green algae symbionts: a tractable model to understand intracellular symbiosis"

_PeerJ, doi:10.7717/peerj.10654_

## Round 0.1 · original submission · Minor Revisions

Thanks for considering PeerJ to submit this very interesting paper.

Overall, the reviews were positive, and the paper and the data presented are novel and important. I understand that the main focus of the paper is the host, however, due to the nature of this work as it is based around symbiosis, I agree with both reviewers that addressing the issue of the algal symbiont is important. While it is true that the genome of the algal symbiont is not yet available there are at least 6 Chlorella genomes available at <https://phycocosm.jgi.doe.gov/phycocosm/home>, including the symbiotic a99 strain. An approach combining matching the RNASeq reads to this collection of chlorella genomes + de novo assembly can really improve the resources availble for this system and address the reviewer’s concerns. Therefore it is important that the authors address this issue by either providing an algal transcriptome or addressing clearly why it wasn’t targeted in this analysis and make changes to the main text to clearly explain why the algal portion of the transcriptome isn’t considered.

In addition, there are some other minor issues mentioned by the reviewers please provide a detailed response highlighting the changes made.

Please in the next draft Make sure to upload all the raw data to the SRA database and make the transcriptome available at NCBI / GigaDB/FigShare or other public repositories.

Reviewer 1 ·

Basic reporting

The authors performed successful infection experiments using a freshwater sponge and their native symbiotic Chlorella algae as a model organism to test hypothesis of interactions and specificity. The authors analyzed the host transcriptomic data to identify key genetic pathways in response to the established symbiosis. In the analysis they used symbiont-free sponges as the control group for the differential expression analysis. The research question is well defined, and the study is relevant to a broad range of scientific community.

Experimental design

General remarks:
The study is missing a key information on the genetic response of the algal symbiont. The authors mentioned that they did not assemble the algae transcriptomic data due to lack of genome references to this particular symbiotic strain. It would be beneficial to the study to perform de novo transcriptome assembly of the symbiont and perform the genes expression comparison between the free-living cultured strain and the symbiotic strain. The AUGUSTUS software can be used to predict genes from protein homology information when mapped to other available Chlorella genomes.
Minor points:
Please specify the method/ kit that was used for preparing the RNA libraries by the LC sciences
Line 222 says that you sequenced with NovaSeq 6000 sequencing system LC Sciences, but line 268-269 says that libraries were sequenced on Illumina HiSeq 4000 platform. Please fix this.
Line 237-238 the sentence is not completed.
In the results the authors are encouraged to include a table with all upregulated and downregulated genes.
The discussion is very lengthy and can be shorten for at least the first few pages, some sections can be moved to the introduction.

Validity of the findings

The authors draw their conclusions based on comparing gene expression values between symbiont-free sponge and sponge re-supplied with cultured algae, 24 hours post-infection. The TEM provide evidence that symbionts colonized the host (Figure 4), however, the authors did not provide a supporting data or information that this 24 hours time point is critical for detecting a genetic response associated with the established symbiosis. The findings in this study would be more supported if gene expression data were evaluated at different time points.

Reviewer 2 ·

Basic reporting

The article entitled "Freshwater sponge hosts and their green algae symbionts: a tractable model to understand intracellular symbiosis" is well written, interesting, and novel transcriptome study shedding light on sponge-Chlorella photosymbiosis.

As pioneers in the transcriptomics of this attractive symbiotic system, I suggest that the authors provide an extra figure(s) with a general model with cellular/metabolic pathways that depict their DEGs/main findings. This will make a stronger point when proposing this photosymbiosis model system.

The figures, specifically the microscopy and pictures provided are excellent. However, figures related to the differential expression analyses need bigger fonts, bigger figures, and self-explanatory labeling.

Finally, the authors should change the Symbiodiniaceae nomenclature at different parts of the paper (see LaJeunesse, et al 2018, Current Biology and the reviewed pdf version).

Experimental design

The authors claimed at line 283: “We do not have a genome sequence available for the sponge-derived Chlorella-like algae and thus were not able to map algal transcripts from the RNASeq data set at this time”. This explanation is insufficient to explain the lack of algal transcriptome analysis, the authors could do a de novo transcriptome assembly and map the algal reads to this assembly. It is understandable that in this paper the authors are not focusing on the algal portion, but this should not be the full explanation, perhaps the authors could solve this issue by simply removing this sentence.

Validity of the findings

The discussion section could be improved when considering the following points:

1. Line 598: Do the authors lookup for glutamate synthase, and ammonium transporters differential gene expression? Include information when possible.

2. Line 600: This part of the discussion needs to be enriched, and the function of asparagine synthetase and its link with nitrogen assimilation/metabolism should be clearly stated.

3. Line 603-611: This part of the discussion could be enriched with the perspectives and future experiments, such as direct manipulation
of the nitrogen sources (amendments) and or carbohydrates such as maltose, to get deeper insights on this.
Do the authors found a carbohydrate/ nitrogen transporter differentially expressed among apo vs. symbiotic sponges?
Could the authors provide a broader comparative approach by using the Hydra- Chlorella genes (carbohydrate/nitrate/ammonium transporters) from Hamada et al., 2018 (or others), and assess their expression in this sponge-Chlorella study? This will only enrich this study and will shed light on the gene expression conservation (if any) when comparing diverse taxa establishing symbioses with this green algae.

Additional comments

The authors will find comments and revisions throughout the pdf file.

Annotated reviews are not available for download in order to protect the identity of reviewers who chose to remain anonymous.

---

## Round 0.2 · Minor Revisions

Thanks for addressing all the reviewer questions, adding the algal component and improving the manuscript overall. I think the manuscript is ready for publication however I am sending it back for minor revisions because I noticed that there might be an inconsistency with regards to figure 6. There is an "extra" legend that does not match the rest of the heatmap where it seems to be listing some GO terms? Which aren't part of this figure. Please address this minor issue and resubmit.

---

## Round 0.3 · accepted · Accept

Thanks for addressing the issues pointed out. The manuscript is ready for publication, congratulations.